# Gaps, challenges and opportunities towards achieving the 95-95-95 targets in Cameroon: A systematic review and meta-analysis protocol

Ezechiel Ngoufack Jagni Semengue[1]*, Alex Durand N. K. A.[1], Pamela Patricia Tueguem[1], Christelle Aude Ka'e[1], Evariste Molimbou[1,2,3], Lum Yah Forgwei[1,4], Naomi-Karell Etame[1], Aurelie Minelle Kengni Ngueko[1,2], Désiré Takou[1], Aristide Stephane Abah Abah[5], Alice Ketchadji[5], Pamen Bouba[6,7], David Kob Same, III[8], Oscar Etogo[9], Basile Keugoung[10], Abdelkader Bacha[10], Phanuel Habimana[9], Martin Samuel Sosso[1], Alexis Ndjolo[1,7], Carlo-Federico Perno[1,11], Vittorio Colizzi[1,3], Rogers Ajeh[12], Anne-Cecile Z-K Bissek[7,13], Nicaise Ndembi[14,15], Joseph Fokam[1,12,16]*

1 Chantal BIYA International Reference Centre for Research on HIV/AIDS Prevention and Management, Yaoundé, Cameroon, 2 Faculty of Medicine and Surgery, University of Rome "Tor Vergata", Rome, Italy, 3 Faculty of Science and Technology, Evangelic University of Cameroon, Bandjoun, Cameroon, 4 School of Health Sciences, Catholic University of Central Africa, Yaoundé, Cameroun, 5 Department of Disease, Epidemics and Pandemic Control, Ministry of Public Health, Yaoundé, Cameroon, 6 Department of Financial resources and assets, Ministry of Public Health, Yaoundé, Cameroon, 7 Faculty of Medicine and Biomedical Sciences, University of Yaoundé I, Yaoundé, Cameroon, 8 Joint United Nations Programme on AIDS (UNAIDS), Country Office, Yaoundé, Cameroon, 9 World Health Organization, Country Office, Yaoundé, Cameroon, 10 United Nations Children's Fund (UNICEF), Country Office, Yaoundé, Cameroon, 11 Bambino Gesu' Children's Research Hospital, Rome, Italy, 12 Faculty of Health Sciences, University of Buea, Buea, Cameroon, 13 Division of Health Operational Research, Ministry of Public Health, Yaoundé, Cameroon, 14 Africa Centres for Disease Control and Prevention, Addis Ababa, Ethiopia, 15 Institute of Human Virology, University of Maryland School of Medicine, Baltimore, MD, United States of America, 16 Central Technical Group, National AIDS Control Committee (NACC), Ministry of Public Health, Yaoundé, Cameroon

* ezechiel.semengue@gmail.com (ENJS); josephfokam@gmail.com (JF)

## Abstract

### Background

Given HIV elimination by 2030, the World Health Organization and the United Nations' Programme on HIV/AIDS (UNAIDS) have set three programmatic goals that all countries should have achieved before 2025; 95% of people living with HIV should know their status (target-1); 95% of people who know their status should be linked to treatment (target-2); and 95% of people on treatment should achieve viral load suppression (i.e. VL<1000 copies/ml; target-3). Despite considerable global progress over the past decade, many low-middle-income countries are still below the expected targets. This review protocol aims to provide a standardized document for in-depth analysis of the strengths, weaknesses, opportunities and threats (SWOT) towards achieving these programmatic targets in Cameroon.

**Data Availability Statement:** Data sharing does not apply to this paper as no datasets were

generated or analyzed during the write-up of the current protocol. The latter is however fully available in the PROSPERO repository (https://www.crd.york.ac.uk/PROSPERO) under the registration number (CRD42024502755).

**Funding:** The author(s) received no specific funding for this work.

**Competing interests:** The authors have declared that no competing interests exist.

**Abbreviations:** AJOL, African Journals Online; ART, antiretroviral therapy; ARV, antiretrovirals; CAHREF, Cameroon Forum Health Research Forum; CENTRAL, Cochrane central register of Trials Controlled Trials; CINAHL, Cumulative Index to Nursing and Allied Health Literature; CROI, conference on retroviruses and opportunistic infections; FDA, food and drug administration; HIV/AIDS, human immunodeficiency virus infection/acquired immune deficiency syndrome; HIVDR, HIV drug resistance; IAC, international aids conference; IAS, international aids society; ICTRP, international clinical trials registry platform; MeSH, medical subject headings; LILAC, latin american and Caribbean health sciences literature; PMTCT, prevention of mother-to-child transmission; PLHIV, people living with HIV; PRISMA-P, preferred reporting items for systematic reviews and meta-analyses protocols; SWOT, strengths, weaknesses, opportunities and threats; UNAIDS, United Nations Programme on HIV/AIDS; WHO, World Health Organization..

## Methods

This systematic review will include randomized and non-randomized trials, cohorts, case-controls, cross-sectional studies, case reports and governmental notices addressing the achievements, the gaps, the challenges and the opportunities towards the 95-95-95 targets in Cameroon. The search will consider studies from 2017 to 2024, retrieved from PubMed/MEDLINE, Cochrane Central Register of Controlled Trials, Google Scholar, online African journals, and Cumulative Index to Nursing and Allied Health Literature. We will include studies reporting HIV diagnosis, HIV link to treatment or HIV viral suppression in Cameroon. Results will be stratified according to time, age-group (adults' vs adolescents/children) and geographic locations. Primary outcomes will be "estimates on 95-95-95 programmatic goals at the national level". Secondary outcomes will consist of the SWOT analysis towards achieving these programmatic targets. A random-effects model will be used to calculate pooled prevalence if data permits.

## Conclusions

This systematic review and meta-analysis protocol will guide global estimation on the achievement of the 95-95-95 targets as well as the stakes and challenges within the Cameroonian setting. Final evidence from the systematic review will allow identification of the gaps and measures to be taken to fasten our move towards the 95-95-95 in Cameroon by 2025.

## Systematic review registration

**Systematic review registration:** CRD42024502755.

## Introduction

The human immunodeficiency virus (HIV) remains a public health problem in every country in the world, with over 20.6 million [18.9 million-23.0 million] people living with HIV (PLHIV) in Eastern and Southern Africa, representing 54% of all PLHIV worldwide. Sub-Saharan Africa remains the region most affected, with over 70% of PLHIV [1, 2]. In Cameroon, HIV prevalence is 2.7% among people aged 15 to 49 and 1.2% among children and adolescents [3, 4]. There is no cure for HIV infection so far. However, thanks to access to effective prevention, diagnosis, treatment and care, including for opportunistic infections, HIV has become a chronic condition that can be managed with the possibility of living a long and healthy life [5, 6]. Africa has made considerable progress in the fight against HIV over the last decade, reducing the number of new infections by 43% and almost halving the number of AIDS-related deaths [7]. However, according to an analysis by the World Health Organization (WHO), it is still "possible" to end the acquired immune deficiency syndrome (AIDS) as a public health threat by 2030. Still, it would be important to improve access to treatment so that people live longer and productive lives [7–9].

To achieve the public health goal of ending the HIV/AIDS epidemic by 2030, the WHO and the United Nations Programme on HIV/AIDS (UNAIDS) have set three (3) targets: 95-95-95, which stipulates that by 2025, all countries should ensure that: 95% of people living with HIV know their serostatus (target-1); 95% of people who know their serostatus are

placed on treatment (target-2); and 95% of people who receive treatment suppress their viral load (target-3) [2, 4, 10–13].

To our knowledge, there is no global synthesis on the achievement of the three 95-95-95 as well as the stakes and the challenges in the Cameroonian context. The regional data on the levels of progress of the 95-95-95 are from now on crucial for a better appreciation/perception of the achievement of these public health objectives. This systematic review and meta-analysis protocol thus aims to provide a standardized document to the scientific community in order to study the global prevalence of the 95-95-95 programmatic goals as well as the stakes and challenges in the Cameroonian context.

## Methods

The present protocol has been registered within the PROSPERO database and is available (on registration number CRD42024502755) and followed the recommendation of the Preferred Reporting Items for Systematic Reviews and Meta-Analyses Protocols (PRISMA-P) statement [14] (see complete PRISMA-P 2015 checklist in S1 File) and for guidelines systematic review protocols (PRISMA-P) (http://www.prisma-statement.org/Extensions/Protocols). PRISMA focuses on ways in which authors can ensure a transparent and complete reporting of systematic reviews and meta-analyses.

### Eligibility criteria

**Design and setting of the study.** *Review question*. What are the strengths, weaknesses, opportunities and threats (SWOT) towards achieving the 95-95-95 targets in Cameroon?

*Type of studies to be included*. We will include randomized and non-randomized trials, cohorts, cross-sectional studies, case reports and governmental notices and reports evaluating the achievements, gaps, challenges and opportunities towards the 95-95-95 targets in Cameroon.

**Characteristics of the participants.** *Participants*. We will include studies focusing on diagnosis/testing, access to test results, patient referral to optimal therapeutic protocols and viral suppression rates; conducted from 2017 to 2024.

*Intervention*. All interventions towards the achievement of any of the 95-95-95 targets will be of interest in this review.

*Comparators*. We will compare the observations in this review and discuss them firstly according to age group and geographic locations in Cameroon; and secondly to the global frame of sub-Saharan Africa at large.

*Outcomes*. Primary outcomes will be "the prevalence of people living with HIV who know their serostatus" in Cameroon; "the prevalence of PLHIV on antiretroviral treatment"; and "the prevalence of PLHIV who have achieved a suppressed viral load in Cameroon. Secondary outcomes will consist of identifying the strengths, weaknesses, opportunities and threats (SWOT) towards achieving the 2025 public health objectives in Cameroon.

The prevalence of people living with HIV who know their serostatus refers to the proportion of seropositive patients who know their serostatus in Cameroon. The prevalence of PLHIV on antiretroviral treatment corresponds to the proportion of PLHIV who know their serostatus who are linked to care and followed up and are on antiretroviral therapy in a health facility. The prevalence of PLHIV who have achieved a suppressed viral load (VL) in Cameroon corresponds to the proportion of PLHIV on antiretrovirals (ARVs) with VL<1000 copies/mL in Cameroon.

*Report characteristics*. We will include studies/reports published in English or French between 2017 and 2024. In effect, UNAIDS programmatic goals were first launched in 2017 as

"the three 90-90-90 goals for 2020" and were later revised as the 95-95-95 targets for 2025. In order to limit translation bias, French articles/reports will be assessed by co-authors whose first-language is French and English articles/report will be assessed by co-authors whose first-language is English.

**Information sources and search strategy.** *Electronic databases.* We will carry out a comprehensive literature search PubMed/MEDLINE, Excerpta Medica (EMBASE), Cochrane Central Register of Controlled Trials (CENTRAL), Science Direct and Cumulative Index to Nursing and Allied Health Literature (CINAHL).

*Trial registers.* Ongoing trials will be sought in the WHO International Clinical Trials Registry Platform (ICTRP) and ClinicalTrials.gov (https://clinicaltrials.gov/).

*Conference abstracts.* We will search conference abstract archives on the websites of the National Forum on Prevention of Mother-to-child Transmission and Management of HIV in Children and Adolescents in Cameroon (PMTCT Forum); and Cameroon Health Research Forum (CAHREF) and all health education conferences, for all available abstracts presented at all conferences from January 2017. We will also search international conference abstract archives on the websites of the Conference on Retroviruses and Opportunistic Infections (CROI); the International AIDS Conference (IAC); and the International AIDS Society Conference on HIV Pathogenesis, Treatment, and Prevention (IAS).

*Other sources.* Hand-searching of the reference lists of relevant reviews and trials will be conducted. In addition, we will contact experts in the field for other potentially eligible studies we may have missed.

The Medical Subject Headings (MeSH terms) for HIV and AIDS and key terms "screening", "ARVs", "viral load suppression", and "Cameroon" (S2 File shows the detailed search strategy for Pubmed, Embase, and CINAHL). We will update the search papers published recently.

### Study records

**Data management.** All documents from the various sources included in our search strategy will be combined and uploaded into the Mendeley reference management software (version 2.83.0). Duplicates will be eliminated from the analysis.

**Selection of eligible studies.** Articles extracted from the databases will be evaluated independently by two authors (LYF and PPT), for title and abstract eligibility. Any disagreements will be resolved by discussion, or consensus, or will involve a third review author (EJNS or ADN) as referee. Two review authors (NKE and PPT) will independently evaluate the full text of eligible papers. Differences will be resolved by consensus or by the arbitration of a third author (ENJS or ADN). The agreement between the first three authors will be estimated by Cohen's kappa coefficient. The PRISMA-P [15] study flow diagram will reflect this process and detail the reasons for the exclusion of studies.

**Data collection.** After checking the eligibility of published articles, a Google questionnaire will be used to extract relevant data and information. Two reviewers will independently read each eligible full-text article and extract the relevant data. Both sets of data will be entered into Microsoft Excel (version 2016 for Windows, Microsoft Corp., Redmond, WA, USA). Any discrepancies in the extracted data will be resolved by consensus, in discussion with a third reviewer (ACK or DT) if necessary.

**Data items.** We will extract the following from the included studies:

- Study characteristics (year of publication, country, study design and study period)

- Characteristics of the study population (sample size, age, sex, inclusion and exclusion criteria)

- PLHIV knowing their status

- PLHIV adherent to ARVs

- ARV regimen

- PLHIV with viral load < 1000 copies/mL

- HIVDR levels

- HIV drug resistance mutations

- HIV subtypes

There is no pre-planned data assumption.

**Data synthesis.** Data analysis will be performed using the "meta" and "metafor" packages of the R statistical software via the RStudio interface (V.4.3.1, R Foundation for Statistical Computing, Vienna, Austria) [16, 17]. Study heterogeneity is estimated using the H statistical test and quantified by the $I^2$ value [18]. The $I^2$ value is used to calculate the percentage of the total variation between studies due to genuine differences between studies rather than chance. The degree of heterogeneity with values of 0%, 18%, 45% and 75% with p < 0.05 will designate zero, low, moderate and high heterogeneity respectively [18], Moderate and high respectively [19]. The prevalence of people living with HIV who know their serostatus in Cameroon and the 95% confidence intervals (95% CI) will be calculated with the "meta-prop" command using a random-effects model [20]. Subgroup analyses will be performed according to study design, undefined geographic area [21], and the proportion of PLHIV adhering to antiretroviral therapy, to adjust for variations in the pooled prevalence estimate. Results will be considered statistically significant if p<0.05. The antiretroviral sensitivity approach will be used to classify the certainty of the evidence as 'high', 'moderate', 'low' and 'very low'.

**Additional analyses.** Subgroup and further analyses will be performed after stratification of study participants. Results will then be sorted according to age (adults/adolescents vs children [<10 years]), PLHIV who know their status, have adhered to ARVs and have a suppressed viral load (<1000 copies/mL). This will enable us to adjust for potential confounding factors, to better estimate the effect of each variable on the observed results. Data permitting, meta-regressions will be performed and summary estimates will be used to explore the relationship between study variables and observed effects, to highlight any statistical significance.

**Dealing with missing data.** If data on key variables are missing, we will contact the authors to obtain clarification of the study. A description of the missing data for each study will be provided and we will discuss the possible implications of the missing data.

**Quality assessment and risk of bias.** Major findings will be summarized in a table. This table will present the quality of the evidence found, all sorted according to socio-demographic data and clinical performances. Though the highest quality rating of the evidence will be for randomized trials, they may be downgraded to moderate, low, or even very low quality based on the risk of bias assessment. Rating will depend on limitations in study design and implementation, indirectness of evidence, unexplained heterogeneity, imprecision of result and a high probability of publication bias [22]. Evidence from observational studies (cohorts and case-control studies) will be graded as low quality [22]. However, if such studies yield large effects and there is no obvious bias explaining those effects, we would rate the evidence as moderate or–if the effect is large enough–even high quality [22]. Detailed interpretation of each evidence and its respective recommendation is provided in S3 File.

The evaluation of included studies for the risk of bias will be done using ROBINS-1 [23, 24], a tool for assessing the risk of bias in non-randomized studies for interventions. ROBIS [RoB 2.0] [23, 24], will be used for randomized controlled trial studies. For observational studies, we will use quality assessment tools for observational cohort and cross-sectional studies [25]. Discrepancy in the risk of bias assessment among the review authors will be solved by discussion and consensus, or by arbitration of a third review author. The publication bias will also be assessed by visual inspection of the asymmetry of the funnel plot and the Egger test with the value of p<0.1 indicating a potential bias [26].

**Meta-biases.** The publication bias will also be assessed by visual inspection of the asymmetry of the funnel plot and the Egger test with the value of p <0.1 indicating a potential bias [26]

**Statistical software.** All analyses will be done in Epi info™ version 7 (CDC, USA) and Microsoft Excel version 2016 (Windows, Microsoft Corp., Redmond, WA, USA). Epi info™ will help us calculate means, medians, frequencies, percentages, and confidence intervals, and assess primary associations between variables using statistical tests. We will use a validated Excel spreadsheet for meta-analysis and forest plots as previously described [27].

## Discussion

This systematic review and meta-analysis will contribute to updating the knowledge on the progress report towards the achievement of the three 95-95-95 programmatic targets and will help to understand the gaps, challenges and opportunities proper to the Cameroonian context. These will thus help to develop new strategies to address the shortcomings and pitfalls towards achieving these public health targets. Findings from this systematic review will inform national and international stakeholders and will help to fasten the efforts and strengthen health system interventions, HIV prevention and treatment strategies for rapid elimination of HIV/AIDS in Cameroon in particular and LMICs in general. As potential limitations of this study protocol, we may be confronted with important study heterogeneity and incompleteness given high variability between study designs, methodologies, participants characteristics; but these will be considered in statistic models during meta-regression analyses; if not performed, study incompleteness at least be solved by contacting the study authors. Availability or quality of data reported may also be limited, and this could affect the robustness of the meta-analysis. Another limitation may be at the level of publication bias, as the review may be influenced by the strict inclusion of published studies only and conference abstracts, which might not represent all relevant data. In effect, in the process of resolving disagreements while reviewing/including articles, all team members' opinions will be considered in the decision-making process or at least they will be aware of the disagreements being discussed. The review might face constraints related to time, resources, or access to certain databases and literature, which could affect the comprehensiveness of the review; variability in quality assessment tools could also limit or affect the interpretation of findings. Furthermore, the protocol might be rigid and not easily adaptable to new evidence or changes in the research landscape. Updates to the protocol or search strategy might be necessary to address emerging data or shifts in the field. However, significant adjustments to the protocol will be documented, taken into consideration during data analysis and discussed accordingly in the final manuscript. The assessment of the quality and risk of bias in studies could be subjective and dependent on the tools used (e.g., ROBINS-I, ROBIS). Variability in assessment could affect the interpretation of the evidence. Finally, the findings of the review will be context-specific to Cameroon and might face constraints related to time, resources, or access to certain databases and literature; this may limit the generalizability of the findings to other regions or countries with different healthcare systems and epidemiological profiles. Our results will therefore be communicated first through national platforms

for local endorsement and also presented at conferences and published in peer-reviewed journals to guarantee broader dissemination of the findings outside the country.

## Supporting information

**S1 File. PRISMA-2015 checklist.**
(DOCX)

**S2 File. Search strategy.**
(DOCX)

**S3 File. Assessing the quality of evidence and the strength of recommendations.**
(DOCX)

## Acknowledgments

The authors are thankful to the "Chantal BIYA" International Reference Centre for Research on HIV/AIDS Prevention and Management (CIRCB), which accepted to support the writing of this work and the subsequent development of the systematic review and meta-analysis.

## Author Contributions

**Conceptualization:** Ezechiel Ngoufack Jagni Semengue, Alex Durand N. K. A, Joseph Fokam.

**Data curation:** Ezechiel Ngoufack Jagni Semengue, Pamela Patricia Tueguem, Evariste Molimbou, Lum Yah Forgwei, Aurelie Minelle Kengni Ngueko, Joseph Fokam.

**Formal analysis:** Ezechiel Ngoufack Jagni Semengue, Pamela Patricia Tueguem, Christelle Aude Ka'e, Lum Yah Forgwei, Naomi-Karell Etame, Aurelie Minelle Kengni Ngueko.

**Investigation:** Ezechiel Ngoufack Jagni Semengue, Alex Durand N. K. A, Pamela Patricia Tueguem, Christelle Aude Ka'e, Evariste Molimbou, Lum Yah Forgwei, Naomi-Karell Etame, Aurelie Minelle Kengni Ngueko.

**Methodology:** Ezechiel Ngoufack Jagni Semengue, Alex Durand N. K. A, Pamela Patricia Tueguem, Christelle Aude Ka'e, Evariste Molimbou, Lum Yah Forgwei, Naomi-Karell Etame.

**Project administration:** Aristide Stephane Abah Abah, Alice Ketchadji, Pamen Bouba, David Kob Same III, Oscar Etogo, Alexis Ndjolo, Carlo-Federico Perno, Vittorio Colizzi, Joseph Fokam.

**Resources:** Désiré Takou, Aristide Stephane Abah Abah, Alice Ketchadji, Pamen Bouba, Oscar Etogo, Basile Keugoung, Abdelkader Bacha, Phanuel Habimana, Martin Samuel Sosso, Alexis Ndjolo, Carlo-Federico Perno, Vittorio Colizzi, Rogers Ajeh, Nicaise Ndembi, Joseph Fokam.

**Software:** Alex Durand N. K. A, Pamela Patricia Tueguem, Christelle Aude Ka'e, Vittorio Colizzi.

**Supervision:** Désiré Takou, Aristide Stephane Abah Abah, David Kob Same III, Phanuel Habimana, Martin Samuel Sosso, Alexis Ndjolo, Carlo-Federico Perno, Anne-Cecile Z-K Bissek, Joseph Fokam.

**Validation:** Désiré Takou, Aristide Stephane Abah Abah, Alice Ketchadji, Pamen Bouba, David Kob Same III, Oscar Etogo, Basile Keugoung, Abdelkader Bacha, Phanuel Habimana,

Alexis Ndjolo, Carlo-Federico Perno, Rogers Ajeh, Anne-Cecile Z-K Bissek,
Nicaise Ndembi, Joseph Fokam.

**Visualization:** Christelle Aude Ka'e, Désiré Takou, David Kob Same III, Oscar Etogo,
Basile Keugoung, Abdelkader Bacha, Phanuel Habimana, Martin Samuel Sosso,
Alexis Ndjolo, Carlo-Federico Perno, Vittorio Colizzi, Anne-Cecile Z-K Bissek,
Nicaise Ndembi, Joseph Fokam.

**Writing – original draft:** Ezechiel Ngoufack Jagni Semengue, Alex Durand N. K. A,
Pamela Patricia Tueguem, Evariste Molimbou, Lum Yah Forgwei, Naomi-Karell Etame,
Aurelie Minelle Kengni Ngueko, Désiré Takou, Aristide Stephane Abah Abah,
Alice Ketchadji, Pamen Bouba, David Kob Same III, Oscar Etogo, Basile Keugoung,
Abdelkader Bacha, Phanuel Habimana, Martin Samuel Sosso, Alexis Ndjolo,
Carlo-Federico Perno, Vittorio Colizzi, Rogers Ajeh, Anne-Cecile Z-K Bissek,
Nicaise Ndembi, Joseph Fokam.

**Writing – review & editing:** Ezechiel Ngoufack Jagni Semengue, Alex Durand N. K. A,
Pamela Patricia Tueguem, Christelle Aude Ka'e, Evariste Molimbou, Lum Yah Forgwei,
Naomi-Karell Etame, Aurelie Minelle Kengni Ngueko, Désiré Takou, Aristide Stephane
Abah Abah, Alice Ketchadji, Pamen Bouba, David Kob Same III, Oscar Etogo,
Basile Keugoung, Abdelkader Bacha, Phanuel Habimana, Martin Samuel Sosso,
Alexis Ndjolo, Carlo-Federico Perno, Vittorio Colizzi, Rogers Ajeh,
Anne-Cecile Z-K Bissek, Nicaise Ndembi, Joseph Fokam.

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
