## [Decision Letter · Decision Letter 0]

4 Jul 2024

PONE-D-24-15119Gaps, challenges and opportunities towards achieving the 95-95-95 targets in Cameroon: A Systematic review and meta-analysis Protocol

PLOS ONE

Dear Dr. Ngoufack Jagni Semengue,

Thank you for submitting your manuscript to PLOS ONE. After careful consideration, we feel that it has merit but does not fully meet PLOS ONE’s publication criteria as it currently stands. Therefore, we invite you to submit a revised version of the manuscript that addresses the points raised during the review process.

Specifically

As data elements are to be captured for 95-95-95, the second 95 is about linkage to ART which is not the right definition for the second 95. The second 95 deals with the proportion of people alive and on treatment. Revise this data element. In the same way, HIV diagnosis is not the right definition for the first 95. Stick with the global definition for 95-95-95.How you plan to do the SWOT analysis is not clearWhat randomized and non-randomized trial types of studies do you expect for this review of 95-95-95? Hence, it does not apply to this analysisAuthors frequently used Diagnosis/Testing for the first 95%, the actual definition is “Know their HIV status” which means “test and result delivered” which means “aware of their status”. That is not about test coverage. Be consistent with the definition of diagnosis/testing which are different in their context. Hence, I strongly suggest you stick with the global definition for the 95-95-95…..Having evidence from studies focusing on diagnosis/testing, access to test results, patient referral to optimal therapeutic protocols, and viral suppression rates; conducted from 2017 to 2024”……, authors can’t get complete information about the 95-95-95. Revise the participants as per the data elements and definition of the 95-95-95……Intervention: all intervention towards the achievement of any of the 95-95-95 …. Qualify what type of interventions would you require to collectGood to finally see the Outcomes of this review which are correct!How do authors plan to handle dual language publications for analysis, do they plan translation of one of the languages?From the data items to be captured I can't see the second 95 (proportion of people on treatment)I still can’t see what data elements will be captured to analyze the SWOT analysisData analysis considers only the first 95%, you don’t have a plan to analyze the second and third 95?==============================

We look forward to receiving your revised manuscript.

Kind regards,

Yimam Getaneh Misganie (PhD, PhD)

Academic Editor

PLOS ONE

Journal Requirements

https://systematicreviewsjournal.biomedcentral.com/articles/10.1186/s13643-020-01356-z

https://doi.org/10.1371/journal.pone.0272839

https://doi.org/10.1002/14651858.CD006495.pub4

In your revision ensure you cite all your sources (including your own works), and quote or rephrase any duplicated text outside the methods section. Further consideration is dependent on these concerns being addressed.

Reviewers' comments:

Reviewer's Responses to Questions

**Comments to the Author**

1. Does the manuscript provide a valid rationale for the proposed study, with clearly identified and justified research questions?

Reviewer #1: Yes

Reviewer #2: No

2. Is the protocol technically sound and planned in a manner that will lead to a meaningful outcome and allow testing the stated hypotheses?

Reviewer #1: Yes

Reviewer #2: No

3. Is the methodology feasible and described in sufficient detail to allow the work to be replicable?

Reviewer #1: Yes

Reviewer #2: No

4. Have the authors described where all data underlying the findings will be made available when the study is complete?

Reviewer #1: No

Reviewer #2: No

5. Is the manuscript presented in an intelligible fashion and written in standard English?

Reviewer #1: Yes

Reviewer #2: No

6. Review Comments to the Author

You may also provide optional suggestions and comments to authors that they might find helpful in planning their study.

Reviewer #1: This paper investigated the manuscript " Gaps, challenges and opportunities towards achieving the 95-95-95 targets in Cameroon: A Systematic review and meta-analysis Protocol"

I have many suggestions for improving the presentation before accepting it for the Journal: PLOS ONE. The following comments are provided for the authors:

1- The authors should make an effort to provide figures and graphs in the manuscript.

2- The authors should compare the results with those of other countries.

3- The authors should mention the national and international organizations that may be interested in the study.

4- The authors should highlight the recommendations.

5- What are the included and excluded criteria in this study?

Reviewer #2: very vaguely written protocol, not at all fit for a systematic review and meta analysis. There is no specific study research question, no specific search criteria.

7. PLOS authors have the option to publish the peer review history of their article (what does this mean?). If published, this will include your full peer review and any attached files.

Reviewer #1: No

Reviewer #2: No

---

## [Author Response · Author response to Decision Letter 0]

8 Aug 2024

RESPONSES TO REVIEWERS

Thank you for submitting your manuscript to PLOS ONE. After careful consideration, we feel that it has merit but does not fully meet PLOS ONE’s publication criteria as it currently stands. Therefore, we invite you to submit a revised version of the manuscript that addresses the points raised during the review process.

Specifically,

• As data elements are to be captured for 95-95-95, the second 95 is about linkage to ART which is not the right definition for the second 95. The second 95 deals with the proportion of people alive and on treatment. Revise this data element. In the same way, HIV diagnosis is not the right definition for the first 95. Stick with the global definition for 95-95-95.

Response 1: We thank the editor for the comment. We have revised accordingly in the text as suggested (see lines 36-37 & 81-83, pages 2-3).

• How you plan to do the SWOT analysis is not clear

Response 2: We thank the editor for the comment. We have provided more explanation on how we plan to do the SWOT analysis in the text as requested (see lines 207-210, page 7).

• What randomized and non-randomized trial types of studies do you expect for this review of 95-95-95? Hence, it does not apply to this analysis

Response 3: We thank the editor for the comment. Indeed, our objective is to include studies based on their focus on at least one of the 95-95-95 targets; so we are expecting to include randomized or non-randomized studies providing figures on either HIV-diagnosis, HIV-treatment or viral suppression in Cameroon. Up-to this point we have successfully retrieved data from one randomized study addressing viral suppression in Cameroon (the NAMSAL study).

• Authors frequently used Diagnosis/Testing for the first 95%, the actual definition is “Know their HIV status” which means “test and result delivered” which means “aware of their status”. That is not about test coverage. Be consistent with the definition of diagnosis/testing which are different in their context. Hence, I strongly suggest you stick with the global definition for the 95-95-95

Response 4: We thank the editor for the comment. We have revised accordingly in the text as suggested (see lines 36-37 & 81-83, pages 2-3).

• …..Having evidence from studies focusing on diagnosis/testing, access to test results, patient referral to optimal therapeutic protocols, and viral suppression rates; conducted from 2017 to 2024”……, authors can’t get complete information about the 95-95-95. Revise the participants as per the data elements and definition of the 95-95-95

Response 5: We thank the editor for the comment. We have brought clarity and revised as follows: “we will include studies focusing on either the proportion of PLHIV who are aware of their status, the proportion of PLHIV under antiretroviral treatment or the proportion of PLHIV achieving viral suppression; conducted from 2017 to 2024” (see lines 107-110, page 4). 

• ……Intervention: all intervention towards the achievement of any of the 95-95-95 …. Qualify what type of interventions would you require to collect

Response 6: We thank the editor for the comment. We have revised as follows: “No intervention will be assessed in this systematic review. Rather, any data highlighting the achievement of any of the 95-95-95 targets will be of interest in this review” (see lines 111-113, page 4).

• Good to finally see the Outcomes of this review which are correct!

Response 7: We thank the editor for the comment.

• How do authors plan to handle dual language publications for analysis, do they plan translation of one of the languages?

Response 8: We thank the editor for the comment. All data extracted will not be translated as we perfectly understand both languages (French and English). However, all the summary and the reporting will be done in English for harmonization purposes. We have added a note addressing this important point (see lines 129-130, page 5).

• From the data items to be captured I can't see the second 95 (proportion of people on treatment)

Response 9: We thank the editor for the comment. We have revised accordingly (see lines 174-176, page 6).

• I still can’t see what data elements will be captured to analyze the SWOT analysis

Response 10: We thank the editor for the comment. We have revised as follows: “… additional data (specific interventions or programmatic actions) will be extracted and grouped to capture the strengths, weaknesses, opportunities and threats for achieving these UNAIDS’ programmatic goals in Cameroon” (see lines 182-184, page 6).

• Data analysis considers only the first 95%, you don’t have a plan to analyze the second and third 95?

Response 11: We thank the editor for the comment. We have revised accordingly (see lines 194-195, pages 6-7). 

Journal Requirements

Response 12: We thank the editor for the comment. We have revised accordingly throughout the manuscript. 

https://systematicreviewsjournal.biomedcentral.com/articles/10.1186/s13643-020-01356-z

https://doi.org/10.1371/journal.pone.0272839

https://doi.org/10.1002/14651858.CD006495.pub4

In your revision ensure you cite all your sources (including your own works), and quote or rephrase any duplicated text outside the methods section. Further consideration is dependent on these concerns being addressed.

Response 13: We thank the editor for the comment. We have revised accordingly in the manuscript. 

Response 14: We thank the editor for the comment. We have revised accordingly as suggested (see line 300, page 10).

---

## [Decision Letter · Decision Letter 1]

27 Aug 2024

PONE-D-24-15119R1Gaps, challenges and opportunities towards achieving the 95-95-95 targets in Cameroon: A Systematic review and meta-analysis Protocol

PLOS ONE

Dear Dr. Ngoufack Jagni Semengue,

Thank you for submitting your manuscript to PLOS ONE. After careful consideration, we feel that it has merit but does not fully meet PLOS ONE’s publication criteria as it currently stands. Therefore, we invite you to submit a revised version of the manuscript that addresses the points raised during the review process.

Academic Editor Comments:

Describe how you plan to handle translation bias from the French articles, in addition to the English.Overall, this is not a systematic review to address SWOT towards 95-95-95, rather this is a protocol to conduct the review hence the objective of this protocol is to provide a standardized document to the scientific community for future use for similar review.Include the potential limitation of this protocol, indicated below, or clarify in the manuscript how you plan to mitigate the issues:**Data Availability**: There may be limitations in the availability or quality of data reported in the included studies, which could affect the ability to perform robust meta-analysis.**Heterogeneity**: High variability between studies in terms of design, methodology, and participant characteristics could impact the synthesis of results and the ability to draw generalizable conclusions.**Publication Bias**: The review may be influenced by the publication bias of including only published studies and conference abstracts, which might not represent all relevant data.**Quality Assessment**: The assessment of the quality and risk of bias in studies could be subjective and dependent on the tools used (e.g., ROBINS-I, ROBIS). Variability in assessment could affect the interpretation of the evidence.**Context-Specific Findings**: The findings of the review will be context-specific to Cameroon, which may not be generalizable to other regions or countries with different healthcare systems and epidemiological profiles.**Protocol Rigidity**: The protocol might be rigid and not easily adaptable to new evidence or changes in the research landscape. Updates to the protocol or search strategy might be necessary to address emerging data or shifts in the field.**Resource Limitations**: The review might face constraints related to time, resources, or access to certain databases and literature, which could affect the comprehensiveness of the review.==============================

We look forward to receiving your revised manuscript.

Kind regards,

Yimam Getaneh Misganie (PhD, PhD)

Academic Editor

PLOS ONE

Journal Requirements:

Additional Editor Comments:

1. Describe how you plan to handle translation bias from the French articles, in addition to the English.

2. Overall, this is not a systematic review to address SWOT towards 95-95-95, rather this is a protocol to conduct the review hence the objective of this protocol is to provide a standardized document to the scientific community for future use to conduct similar review.

3. Include the potential limitation of this protocol, indicated below, or clarify in the manuscript how you plan to mitigate the issues:

• Data Availability: There may be limitations in the availability or quality of data reported in the included studies, which could affect the ability to perform robust meta-analysis.

• Heterogeneity: High variability between studies in terms of design, methodology, and participant characteristics could impact the synthesis of results and the ability to draw generalizable conclusions.

• Publication Bias: The review may be influenced by the publication bias of including only published studies and conference abstracts, which might not represent all relevant data.

• Quality Assessment: The assessment of the quality and risk of bias in studies could be subjective and dependent on the tools used (e.g., ROBINS-I, ROBIS). Variability in assessment could affect the interpretation of the evidence.

• Context-Specific Findings: The findings of the review will be context-specific to Cameroon, which may not be generalizable to other regions or countries with different healthcare systems and epidemiological profiles.

• Protocol Rigidity: The protocol might be rigid and not easily adaptable to new evidence or changes in the research landscape. Updates to the protocol or search strategy might be necessary to address emerging data or shifts in the field.

• Resource Limitations: The review might face constraints related to time, resources, or access to certain databases and literature, which could affect the comprehensiveness of the review.

Reviewers' comments:

Reviewer's Responses to Questions

**Comments to the Author**

1. Does the manuscript provide a valid rationale for the proposed study, with clearly identified and justified research questions?

Reviewer #3: Yes

2. Is the protocol technically sound and planned in a manner that will lead to a meaningful outcome and allow testing the stated hypotheses?

Reviewer #3: Yes

3. Is the methodology feasible and described in sufficient detail to allow the work to be replicable?

Reviewer #3: Yes

4. Have the authors described where all data underlying the findings will be made available when the study is complete?

Reviewer #3: Yes

5. Is the manuscript presented in an intelligible fashion and written in standard English?

Reviewer #3: No

6. Review Comments to the Author

You may also provide optional suggestions and comments to authors that they might find helpful in planning their study.

Reviewer #3: The authors should make sure that the manuscript is written in standard English before it is accepted for publication. Authors should justify the time frame of their review and add the review question.

7. PLOS authors have the option to publish the peer review history of their article (what does this mean?). If published, this will include your full peer review and any attached files.

Reviewer #3: No

---

## [Author Response · Author response to Decision Letter 1]

12 Sep 2024

Responses to reviewers:

Academic Editor Comments:

1. Describe how you plan to handle translation bias from the French articles, in addition to the English.

Response 1: We thank the Editor for this comment. However, there will not be any translation bias from English to French articles/reports as both languages are adopted at national level and are perfectly mastered by all co-authors.

2. Overall, this is not a systematic review to address SWOT towards 95-95-95, rather this is a protocol to conduct the review hence the objective of this protocol is to provide a standardized document to the scientific community for future use for similar review.

Response 2: We thank the Editor for this comment. We have revised and clarify the aim of this protocol accordingly.

3. Include the potential limitation of this protocol, indicated below, or clarify in the manuscript how you plan to mitigate the issues:

• Data Availability: There may be limitations in the availability or quality of data reported in the included studies, which could affect the ability to perform robust meta-analysis.

• Heterogeneity: High variability between studies in terms of design, methodology, and participant characteristics could impact the synthesis of results and the ability to draw generalizable conclusions.

• Publication Bias: The review may be influenced by the publication bias of including only published studies and conference abstracts, which might not represent all relevant data.

• Quality Assessment: The assessment of the quality and risk of bias in studies could be subjective and dependent on the tools used (e.g., ROBINS-I, ROBIS). Variability in assessment could affect the interpretation of the evidence.

• Context-Specific Findings: The findings of the review will be context-specific to Cameroon, which may not be generalizable to other regions or countries with different healthcare systems and epidemiological profiles.

• Protocol Rigidity: The protocol might be rigid and not easily adaptable to new evidence or changes in the research landscape. Updates to the protocol or search strategy might be necessary to address emerging data or shifts in the field.

• Resource Limitations: The review might face constraints related to time, resources, or access to certain databases and literature, which could affect the comprehensiveness of the review.

Response 3: We thank the Editor for this comment. We have revised our protocol limitation exactly as suggested.

---

## [Editor Report · Decision Letter 2]

24 Sep 2024

PONE-D-24-15119R2Gaps, challenges and opportunities towards achieving the 95-95-95 targets in Cameroon: A Systematic review and meta-analysis ProtocolPLOS ONE

Dear Dr. Ngoufack Jagni Semengue,

Thank you for submitting your manuscript to PLOS ONE. After careful consideration, we feel that it has merit but does not fully meet PLOS ONE’s publication criteria as it currently stands. Therefore, we invite you to submit a revised version of the manuscript that addresses the points raised during the review process.

The current version of the manuscript contains substantial changes that extend beyond addressing the reviewers' and editor's comments from the previous version. As this manuscript has already undergone two rounds of review, the authors are requested to focus strictly on the feedback provided by the reviewers and editor. The decision has been changed from "minor revision" to "major revision" due to the emergence of new issues during the review process, resulting from the revisions made by the authors, which brought to light concerns that were not initially apparent.

We look forward to receiving your revised manuscript.

Kind regards,

Yimam Getaneh (MSc, PhD, PhD)

Academic Editor

PLOS ONE

---

## [Author Response · Author response to Decision Letter 2]

7 Nov 2024

Responses to Editor’s Comments:

Academic Editor Comments:

1. Describe how you plan to handle translation bias from the French articles, in addition to the English.

Response 1: We thank the Editor for this comment. However, there will not be any translation bias from English to French articles/reports as both languages are adopted at national level and are perfectly mastered by all co-authors. In effect some authors have English as first-language for expression whereas others have French as first-language for expression. We have added a comment on that light (see lines 130-132, page 5).

2. Overall, this is not a systematic review to address SWOT towards 95-95-95, rather this is a protocol to conduct the review hence the objective of this protocol is to provide a standardized document to the scientific community for future use for similar review.

Response 2: We thank the Editor for this comment. We have revised and clarify the aim of this protocol accordingly (see line 39, page 2; and lines 88-89, page 3).

3. Include the potential limitation of this protocol, indicated below, or clarify in the manuscript how you plan to mitigate the issues:

• Data Availability: There may be limitations in the availability or quality of data reported in the included studies, which could affect the ability to perform robust meta-analysis.

• Heterogeneity: High variability between studies in terms of design, methodology, and participant characteristics could impact the synthesis of results and the ability to draw generalizable conclusions.

• Publication Bias: The review may be influenced by the publication bias of including only published studies and conference abstracts, which might not represent all relevant data.

• Quality Assessment: The assessment of the quality and risk of bias in studies could be subjective and dependent on the tools used (e.g., ROBINS-I, ROBIS). Variability in assessment could affect the interpretation of the evidence.

• Context-Specific Findings: The findings of the review will be context-specific to Cameroon, which may not be generalizable to other regions or countries with different healthcare systems and epidemiological profiles.

• Protocol Rigidity: The protocol might be rigid and not easily adaptable to new evidence or changes in the research landscape. Updates to the protocol or search strategy might be necessary to address emerging data or shifts in the field.

• Resource Limitations: The review might face constraints related to time, resources, or access to certain databases and literature, which could affect the comprehensiveness of the review.

Response 3: We thank the Editor for this comment. We have revised our protocol limitation exactly as recommended (see lines 247-271, page 8).

Once again, we thank the Editor for the constructive remarks and are very much confident that the current version is suitable for publication. All other comments in the manuscript have also been addressed accordingly (see lines 104-105, page 4; and lines 129-130, page 5).

---

## [Editor Report · Decision Letter 3]

7 Jan 2025

Gaps, challenges and opportunities towards achieving the 95-95-95 targets in Cameroon: A Systematic review and meta-analysis Protocol

PONE-D-24-15119R3

Dear Dr. Ezechiel,

We’re pleased to inform you that your manuscript has been judged scientifically suitable for publication and will be formally accepted for publication once it meets all outstanding technical requirements.

Kind regards,

Yimam Getaneh (PhD, PhD)

Academic Editor

PLOS ONE
---

## [Editor Report · Acceptance letter]

16 Jan 2025

PONE-D-24-15119R3 

PLOS ONE

Dear Dr. Ngoufack Jagni Semengue, 

I'm pleased to inform you that your manuscript has been deemed suitable for publication in PLOS ONE. Congratulations! Your manuscript is now being handed over to our production team.

Kind regards, 

on behalf of

Dr. Yimam Getaneh Misganie 

Academic Editor

PLOS ONE